# On the Risks of Phylogeny-Based Strain Prioritization for Drug Discovery: *Streptomyces lunaelactis* as a Case Study

**DOI:** 10.3390/biom10071027

**Published:** 2020-07-10

**Authors:** Loïc Martinet, Aymeric Naômé, Dominique Baiwir, Edwin De Pauw, Gabriel Mazzucchelli, Sébastien Rigali

**Affiliations:** 1InBioS—Centre for Protein Engineering, Institut de Chimie B6a, University of Liège, B-4000 Liège, Belgium; loic@hedera22.com; 2Hedera-22, Boulevard du Rectorat 27b, B-4000 Liège, Belgium; aymeric@hedera22.com; 3GIGA Proteomics Facility, University of Liège, B-4000 Liège, Belgium; D.Baiwir@uliege.be; 4MolSys Research Unit, Mass Spectrometry Laboratory, University of Liège, B-4000 Liège, Belgium; e.depauw@ulg.ac.be (E.D.P.); gabriel.mazzucchelli@ulg.ac.be (G.M.)

**Keywords:** strain prioritization, metabolomics, chemical diversity, natural product, drug discovery, genome mining, streptomyces

## Abstract

Strain prioritization for drug discovery aims at excluding redundant strains of a collection in order to limit the repetitive identification of the same molecules. In this work, we wanted to estimate what can be unexploited in terms of the amount, diversity, and novelty of compounds if the search is focused on only one single representative strain of a species, taking *Streptomyces lunaelactis* as a model. For this purpose, we selected 18 *S. lunaelactis* strains taxonomically clustered with the archetype strain *S. lunaelactis* MM109^T^. Genome mining of all *S. lunaelactis* isolated from the same cave revealed that 54% of the 42 biosynthetic gene clusters (BGCs) are strain specific, and five BGCs are not present in the reference strain MM109^T^. In addition, even when a BGC is conserved in all strains such as the *bag*/*fev* cluster involved in bagremycin and ferroverdin production, the compounds produced highly differ between the strains and previously unreported compounds are not produced by the archetype MM109^T^. Moreover, metabolomic pattern analysis uncovered important profile heterogeneity, confirming that identical BGC predisposition between two strains does not automatically imply chemical uniformity. In conclusion, trying to avoid strain redundancy based on phylogeny and genome mining information alone can compromise the discovery of new natural products and might prevent the exploitation of the best naturally engineered producers of specific molecules.

## 1. Introduction

Natural products (NPs) display a remarkable array of chemical structures and bioactivities. The current urge for discovering new drug leads in order to face resistance phenomena have boosted bioprospecting in order to access the rare and dark microbial matter [1,2]. As a result, collections of microorganisms multiply, both in academia and in industries, differing by their size (number of strains), their richness (diversity, originality, and rarity of isolated strains), the geographical location of their isolation sites, and the characteristics of the ecological niches prospected.

The number of strains in certain private collections can reach hundreds of thousands of isolates, which, first, limits the number of culture conditions to assess the potential of each isolate to produce NPs and, second, prevents exhaustive investigations despite the implementation of high-throughput cultivation systems and the automation of downstream processes. The tendency is therefore to limit the number of strains to be investigated via prioritization strategies (identical strain dereplication) either based on the phylogeny and other genomic information (when available), or via analyses of chemical profiles and expression patterns [3,4,5,6,7].

In contrast, several works stressed the importance of screening multiple strains of the same species for NP discovery, revealing that taxonomically identical species do not display identical metabolomic patterns [8,9,10]. However, it is actually difficult to estimate how many of these studies have really assessed the metabolite patterns at the “real” strain level. This is due to the difficulty of precisely defining the notion of subspecies in *Streptomyces* and other important genera of “NP-makers”. An identical 16S rRNA sequence is definitely not sufficient enough for species demarcation and can lead to ill-defined species [11,12]. In earlier investigations, Seipke RF performed such an analysis on six *Streptomyces albus* strains, revealing that only 18 of a total of 48 biosynthetic gene clusters were conserved between all strains, thereby highlighting the high number of strain-specific secondary metabolites [9]. However, genome assemblies of some *S. albus* strains contain many contigs on which some biosynthetic gene clusters (BGCs) might be scattered, thereby resulting in an overestimation of the total number of BGCs [9]. In a more recent study, Tidjani AR and colleagues showed, by comparative genomics of *Streptomyces* strains belonging to the same species and isolated at the microscale, that almost-clonal strains can present important genetic content diversity providing them with unique metabolite production capabilities [13].

The aim of this work is to provide a novel case study including both genome mining and metabolomic analysis to illustrate the risks of a strain prioritization strategy. Taking our collection of 18 *Streptomyces lunaelactis* strains isolated from the same cave moonmilk deposits [14,15,16], we showed that, even when phylogeny analyses have demonstrated that multiple strains belong to one single species, the compounds produced can still highly differ in terms of quantity (from basically nothing to economically viable production yields), diversity (structural forms only produced in one or few subspecies), and novelty (sometimes novel compounds are only produced by one single strain).

## 2. Materials and Methods 

### 2.1. Bacterial Strains and Culture Conditions

All *S. lunaelactis* strains used in this study were isolated from moonmilk deposits of the “Grotte des Collemboles” (Comblain-au-Pont, Belgium) [14,15,16]. The R2YE medium—with or without 1 mM FeCl_3_ [17]—was used for measuring the production of ferroverdins. The ISP7 medium was used for measuring the production of bagremycins. Mycelia from liquid ISP1 two-day pre-cultures were used to inoculate solid ISP7 or R2YE media. *Streptomyces* cultivation conditions and spore preparation were performed as described in [18]. *S. lunaelactis* strains used in this study are MM15, MM22, MM25, MM28, MM29, MM31, MM37, MM40, MM51, MM78, MM83, MM91, MM103, MM109, MM113, MM115, MM126, and MMun143. Strains are only designated by their numbers when space is limited in a figure.

### 2.2. Ferroverdin A Quantification by High Pressure Liquid Chromatography (HPLC) 

Ferroverdin extraction with ethyl acetate, drying, and resuspension in acetonitrile were performed as described previously [19]. Samples were analyzed by HPLC and Ferroverdin A was detected at 440 nm and quantified by peak area integration, as described previously [19]. Data were analyzed using Empower 3 (Waters, Milford, MA, USA). The HPLC-based protocol for ferroverdin A semi-quantitative analysis is detailed in Appendix B.

### 2.3. Compound Identification by Ultra-Performance Liquid Chromatography–Tandem Mass Spectrometry (UPLC–MS/MS)

Ferroverdin and Bagremycins compounds in extracts of *S. lunaelactis* strains were analyzed by Ultra-Performance Liquid Chromatography–Tandem Mass Spectrometry (UPLC–MS/MS) following the Ultra-Performance Liquid Chromatography-High Resolution Mass Spectrometry (UPLC–HRMS) method detailed in Appendix C. Each compound was identified according to its exact mass, the isotopic pattern, the MS/MS spectra of the molecular ion HCD fragmentation, and the UV-VIS absorbance spectra.

### 2.4. Molecular Network Construction

MS and MS/MS raw data obtained via the Excalibur Software (Thermo) were converted to a 32-bit mzXML file using MSconvert software (Proteo Wizard toolkit [20]). mzXML files were subjected to the Global Natural Product Social Molecular Networking (GNPS) site (https://gnps.ucsd.edu) as described in [21]. For the parent MS, the mass tolerance was set to 0.01 Da and for the MS/MS fragment ions, the mass tolerance was set to 0.5 Da, while the minimum cosine score was set to 0.7. The data were clustered using MSCluster with a minimum cluster size of four spectra. The spectra in the network were also searched against GNPS spectral libraries. A minimum score of 0.5 was set for spectral library search, with at least two fragment peaks matching. Cytoscape 3.7.1 was used for visualization of the generated molecular networks [22,23]. The edge thickness was set to represent the cosine score, with thicker lines indicating higher similarity between nodes. The Molecular Networking job in GNPS can be found at https://gnps.ucsd.edu/ProteoSAFe/status.jsp?task=84ea45db68814c94a3b2bad9c26dd807. 

### 2.5. Phylogeny Analyses

The Multi Locus Sequence Analysis (MLSA) based on housekeeping genes *atpD*, *gyrB*, *recA*, *rpoB*, and *trpB* of all *S. lunaelactis* strains (Figure 1) was performed as follows. The five genes were identified in the *S. lunaelactis* genomes based on sequence homology, with the loci available from the *Streptomyces* MLST website (https://pubmlst.org/strteptomyces/). For each strain, the 100 closest relative strains (amongst all the available genomes of actinobacteria in NCBI GenBank as of September 2019) were retrieved based on the best cumulative blastp scores of the five housekeeping gene products. From the resulting unique closest relatives, only those with an assembly status of “complete genome” or those tagged as “representative genome” were retained. A protein multiple sequence alignment was performed for each gene (MAFFT v7.453 [24], options—maxiterate 1000—localpair;) as a guide for subsequent nucleotide alignment (PAL2NAL v14 [25]). The five full-length nucleotide sequence alignments were then merged and trimmed (trimAl v1.2rev59 [26], method—automated1). Phylogenetic inference was deduced with the maximum likelihood method, as implemented in RAxML (v8.1.17 [27], rapid bootstrapping mode with 1000 replicates, GTR+I+G evolutionary model), using *S. coelicolor* as the outgroup. The phylogeny analysis based on the full-length *bag*/*fev* cluster of all *S. lunaelactis* strains was performed likewise with the *fev* cluster of *Streptomyces* sp. WK-5344 (1-18563 nt of GenBank accession AB689797.1) as the outgroup. Data for the average nucleotide identity (ANI) of the 18 *S. lunaelactis* strains genomes were generated with the default “compare” workflow of dRep [28] applied to the 18 genomes, implementing the fast MASH [29] and accurate ANIm algorithms [30]. The clustering was performed, and the plot generated with the pheatmap R package using the default hierarchical clustering method (Euclidean distance and complete linkage algorithm).

### 2.6. Genome Sequencing

The genome sequences of all *S. lunaelactis* strains were obtained as described previously [14,15,31]. Genomic DNA was extracted with the GenElute Bacterial Genomic DNA Kit (Sigma-Aldrich, St. Louis, MO, USA). Genomic libraries were constructed using the Nextera XT kit (Illumina, Inc., San Diego, CA, USA). Sequencing was carried out on an Illumina MiSeq platform with 2 × 300-bp read configuration. Complete genomes were assembled de novo from raw sequence data with SPAdes v3.6.2 [32], and the quality of the assemblies was subsequently assessed with QUAST v2.3 [33]. All assemblies are available under NCBI BioProject accession PRJNA30192.

## 3. Results

### 3.1. Genome Mining of 18 S. lunaelactis Strains Revealed Strain-Specific Secondary Metabolism

Previous 16S rRNA-based phylogeny analyses suggested that 18 strains collected during our various campaigns aimed at isolating actinomycetes from karstic environments belong to the species *Streptomyces lunaelactis* [13,14,15]. However, 16S rRNA is a weak marker for species determination in streptomycetes and we therefore performed MLSA using concatenated full-length sequences of housekeeping genes *atpD*, *gyrB*, *recA*, *rpoB*, and *trpB*. As shown in Figure 1, all 18 *Streptomyces* strains are separated from their closest phylogenetic counterparts and cluster together with the archetype strain *S. lunaelactis* MM109^T^ [16,31]. A calculation of the average nucleotide identity percentages (ranging from 98% to 100%) between all 18 isolates confirmed that they belonged to the same species (Appendix A).

A common feature of *S. lunaelactis* strains is the greenish pigmentation of the mycelium resulting from ferroverdin production when cultured on iron-containing media [16,19]. The BGC responsible for ferroverdin biosynthesis (*bag*/*fev*) is also required for the production of bagremycin antibiotics when iron concentrations are less abundant [19]. Genome mining revealed that all 18 *S. lunaelactis* strains possess the *bag*/*fev* BGC (Figure 2, cluster network #12).

The phylogeny analyses based on the 16S rRNA and other housekeeping genes, combined to the presence the *bag*/*fev* cluster, suggest that this subset of our collection unambiguously taxonomically belong to the species *S. lunaelactis* and could be subjected to strain dereplication/prioritization for NP discovery. However, genome mining revealed that only 18 of the 37 BGCs of the type strain *S. lunaelactis* MM109^T^ are conserved in the other 17 *S. lunaelactis* isolates (Figure 2). Five BGCs that belong to metabolite classes NRPS (BGC#28b, BGC#31), nucleosides (BGC#32), T1PKS (BGC#29), and hybrid T1PKS-NRPS (BGC#30), are absent in the type strain MM109^T^ which thus increases to 42 the number of different clusters associated with *S. lunaelactis* species (Figure 2). Two of these additional BGCs are extremely rare as they are exclusively specific to one single *S. lunaelactis* strain, i.e., the NRPS BGC#31 in MM15, and the nucleoside-type BGC#32 in strain MM37. From the 18 conserved clusters, seven of them belong to the so-called the core metabolome [10,26], i.e., BGCs involved in the biosynthesis of molecules produced by almost all *Streptomyces* species (geosmin #4a, ectoin #8, deferoxamine #11b, spore pigment #18, hopene #22, melanin #23c, and alkyresorcinol #26b). Six clusters are only found in strain MM109^T^ and the closely related strains MM37 and MM103 (RiPPs BGC#6, the NRPS BGC#2, 13, 28, terpene BGC#1, and also BGC#19 predicted to be involved in annimycin production).

The specialized metabolism of the species *S. lunaelactis* is therefore more diverse than previously predicted from the genome mining of its type strain MM109^T^. Eighteen BGCs (46%) constitute the conserved secondary metabolome of *S. lunaelactis* strain and five BGCs would not have been identified if strain prioritization would have selected for NP discovery the first characterized strain (MM109^T^). However, it has to be noted that the first isolated strain MM109^T^ outgroups from most other *S. lunaelactis* strains and harbors six BGCs that are not found in most other representatives of this species.

### 3.2. Strain-Specific Compound Diversity and Uneven Levels of Bagremycin and Ferroverdin Production Amongst the Various S. lunaelactis Strains

Albeit a BGC is conserved in all strains of one species, it does not exclude strain-specific mutations that would drastically modify the production level and/or the diversity of the biosynthesized compounds. Phylogeny analysis of *S. lunaelactis* strains based on the nucleotide sequence of the *bag*/*fev* BGC revealed that the reference strain MM109^T^ does not cluster with most of the other *S. lunaelactis* strain isolated from the same moonmilk deposit (Figure 3). 

Comparative analysis of the coding sequence of the 16 genes that compose this cluster revealed a total of 67 different mutations (66 non-silent mutations and one deletion), strain MM51 being the most distant isolate compared to the type strain MM109^T^ cumulating 53 mutations (Figure 4). The gene *fevT* encoding a LuxR-family transcriptional regulator is the open reading frame that contains the highest number of mutations (eleven) compared to the sequence originally found in MM109^T^ (GenBank: AUG90788.1). At the opposite, genes *fevV* (tyrosine ammonia-lyase) and *fevF (o*-aminophenol oxidase) present identical amino acid sequences in the eighteen *S. lunaelactis* strains. MM103 is the *S. lunaelactis* strain closest to MM109^T^ with only two mutations both in *fevL* involved in the decarboxylation of the *trans*-coumaric acid in *p*-vinylphenol (Figure 4). The highlighted sequence heterogeneity suggests that these different strains, despite that they all taxonomically belong to the species *S. lunaelactis*, could indeed present important differences in production levels of both bagremycins and ferroverdins.

### 3.3. Heterogeneous Production of Bagremycins by S. lunaelactis Strains

Bagremycins are amino-aromatic antibiotics predicted to result from the condensation of 3-amino-4-hydroxybenzoic acid with *p*-vinylphenol [19,34]. Previous studies have identified six different bagremycins (bagremycin A to G) produced by species *Streptomyces* sp. Tü 4128 [34], *Streptomyces* sp. Q22 [35], *Streptomyces* sp. ZZ745 [36], and *Streptomyces lunaelactis* MM109^T^ [19]. In addition to their antibacterial and antifungal activities [17,27,29], bagremycins C and F, that differ from the four other bagremycins by the presence of a *N*-acetyl-(*S*)-cysteine moiety, have also been reported to possess an anticancer activity [35]. Finally, bagrelactone A is a bagremycin-derived macrolide isolated from *Streptomyces* sp. Q22 [35].

We previously showed that strain *S. lunaelactis* MM109^T^ is able to produce all known bagremycins except bagremycin D [19] and in this work we first wanted to assess if the other strains of *S. lunaelactis* had similar bagremycin production profiles. Seven *S. lunaelactis* strains were inoculated in the ISP7 medium and bagremycins produced were identified by UPLC–MS/MS as described previously [19]. Molecular ion species corresponding to bagremycins were semi-quantified by peak integration of extracted ion chromatograms (EIC), and levels produced by the type strain MM109^T^ were fixed to 100% for comparative analysis with other *S. lunaelactis* strains. 

As shown in Figure 5, the various bagremycins have different best producing strain, i.e., MM83 for bagremycin A and bagremycin E, MM113 for bagremycins B and bagremycin G, and MM37 for bagremycin C and bagremycin F. This means that the reference strain MM109^T^, used as the model strain for NP studies in *S. lunaelactis*, is never the best producer of any known bagremycin. Access to these alternative bagremycin-producing strains could also solve problems associated with too weak production yields (or in some cases the compound not being produced at all) and facilitate downstream purification steps. For instance, MM83 would be the optimal *S. lunaelactis* strain to use in order to obtain all bagremycin-like compounds. Strain MM37 would be, instead, recommended for extracts enriched in sulfured-bagremycins C and F for utilization in anticancer activity tests. The type strain MM109^T^ would prevent contamination by the most abundantly produced bagremycin (bagremycin E, Figure 5) and would be ideal for bagrelactone production (Figure 5). These results demonstrate that a strain prioritization strategy based on genome mining and phylogeny would have prevented the exploitation of the best naturally engineered producers of specific forms of known bagremycins.

Next to the evaluation of the production levels of known bagremycins, we assessed if the different *S. lunaelactis* strains were able to produce new variants of bagremycins. MS/MS fragmentation analysis of UPLC–HRMS data allowed us to identify three “tag fragments” (Appendix A) associated with the currently known bagremycins. The first ion “tag fragment” of *m*/*z* 121.06 (C_8_H_8_O^+^) corresponds to the *p*-vinylphenol ionized part of all the “group 1” bagremycins (bagremycins A, B, E and G). The sulfur-containing bagremycins (bagremycins C and F, group 2) present another “tag fragment” of *m*/*z* 255.04 (C_10_H_11_N_2_O_4_S^+^). Finally, bagrelactone-like compounds (group 3) are characterized by the “tag fragment” of *m*/*z* 178.05 (C_9_H_8_NO_3_^+^).

To have an overview of all the bagremycin-related metabolites produced by seven *S. lunaelactis* strains (MM25, MM31, MM37, MM40, MM83, MM109, and MM113), the search of these three “tag fragments” in the MS/MS spectra of all UPLC–HRMS analyses was performed to identify bagremycin compounds. In addition to the manual “tag fragment” screening, a Molecular Networking (MN) analysis was carried out by the Global Natural Product Social (GNPS) platform [21] to automatically detect structural relatedness among molecules using the MS/MS data, and molecular connection networks were visualized using Cytoscape [21]. The resulting molecular network of the seven selected *S. lunaelactis* strains contains 977 nodes clustered in 150 constellations (Figure 6). 

The largest network is composed of 66 nodes and contains the tag fragment of bagremycins of Group 1 (fragment tag of *m*/*z* 121.06, as proposed in Appendix A). This cluster indeed includes ion species *m*/*z* 256, 298, 241, 284, corresponding to bagremycin A, bagremycin B, bagremycin E, and bagremycin G, respectively (Figure 6, Group 1). This constellation of 66 nodes also includes the ion *m*/*z* 272 corresponding to the bagrelactone (Group 3, fragment tag of *m*/*z* 178 as proposed in Appendix A). The fact that the Group 1 and Group 3 compounds are included in the same constellation could be explained by the presence of the *m*/*z* 254 ion fragment corresponding to the bagremycin A molecular ion (minus two hydrogen atom) which is also present in the fragmentation pattern of the bagremycin B and the bagrelactone. Finally, another constellation contains 19 ion species (nodes) that share the tag fragment of *m*/*z* 255 specific of Group 2 which includes the sulfur-containing bagremycins as confirmed by the presence of the 417 *m*/*z* ion which corresponds to bagremycin C (Figure 6, Group 2).

With two constellations of 66 (Group 1 and Group 3) and 19 (Group 2) ion species, the MN directly suggests that the chemical diversity of bagremycin-like metabolites produced by *S. lunaelactis* strains is much more important than currently known (only seven bagremycins and one bagrelactone identified so far). A combination of MS/MS fragmentation data with HRMS exact mass and the isotopic distribution of each compound allowed us to propose a list of predicted molecular formulas (Table 1) and to propose the structure for some of the *m*/*z* ions identified by the GNPS software (Appendix A). Next to the identification of the five already known bagremycins (compounds 1 to 5 in Table 1) and the bagrelactone (compound 12, Table 1), Appendix A present, respectively, the MS/MS spectra and molecular fragmentation pathways associated to the UPLC–HRMS/MS data that allowed us to propose a possible structure for six new bagremycins (compounds 6 to 11, Table 1), and four new bagrelactones (compounds 13 to 16, Table 1).

In the constellation of 66 nodes, we identified three new typical bagremycins of Group 1, namely, bagremycin H (*m*/*z* 271), bagremycin I (*m*/*z* 266), and bagremycin J (*m*/*z* 286), for which we propose a plausible molecular structure (Appendix A, Table 1). In the constellation with 19 nodes (Group 2), the ions of *m*/*z* 315 (bagremycin C2), 427 (Bagremycin K), and 441 (Bagremycin L) were identified as new sulfured-bagremycins for which MS/MS fragmentation allowed us to predict a possible structure (Appendix A, Table 1). The constellation of 66 nodes also possess seven ions that belong to Group 3 (bagrelactones) amongst which the three ions of *m*/*z* 296, 270, 302, and 316 were correlated to new bagrelactone B, bagrelactone C, bagrelactone D, and bagrelactone E, respectively (Appendix A and Table 1).

For other nodes, it was possible to accurately propose their chemical formula, but the MS/MS fragmentation data did not allow us to predict a structure (compounds 17 to 33 in Table 1). Amongst these, six nodes are predicted to be other new “classical” bagremycins (compounds 17 to 22, Table 1), and 11 nodes are correlated to sulfured-bagremycins (compounds 23 to 33, Table 1).

In conclusion, as previously observed for the known compounds (Figure 5), the new bagremycins and the new bagrelactones are preferentially produced by certain strains of *S. lunaelactis*, especially strain MM83, while others revealed to be extremely weak producers (strain MM25 only produces three of the 10 bagremycins and none of the five bagrelactones) (Figure 7). From the ten new bagremycin-like compounds identified in this study, only three were detected in the culture extract of the reference strain MM109^T^, further demonstrating how a strain prioritization strategy would have prevented the discovery of these new NPs.

### 3.4. Heterogeneous Production Levels of Ferroverdins by S. lunaelactis Strains

The *fev*/*bag* cluster being also responsible for ferroverdin production, we also assessed how the production of these metabolites was impacted in different *S. lunaelactis* strains. As shown in Figure 8a, seven of the eight selected *S. lunaelactis* strains have conserved the ability to trigger the production of green-pigmented ferroverdins upon the sensing of iron overload. MM91 lost this ability as a result of the inframe deletion identified at position +337 nt of the coding sequence of the transcriptional activator *fevR*, as reported previously [17]. Ferroverdin A is the most abundant ferroverdin in the type strain *S. lunaelactis* MM109 [19], and its production was measured in the eight selected *S. lunaelactis* strains. Like stated earlier for bagremycin and bagrelactone production, strain MM37 was also revealed to be the best producer of the *S. lunaelactis* isolates, with about 3.5 times more production of ferroverdin A compared to the type strain MM109^T^ (Figure 8b).

Aside of ferroverdin A, we investigated the production levels of all ferroverdin-related compounds by Molecular Networking as performed earlier for bagremycins and bagrelactones (Figure 6). This time, we performed MN for only two strains, namely the reference strain MM109^T^ for which ferroverdin production properties have already been described [19], and strain MM37 which is the best ferroverdin producer suggesting that its *bag*/*fev* cluster would be particularly active. After manual curation, all ferroverdin-related molecules in the extracts of both *S. lunaelactis* strains were retrieved to generate a molecular network of 41 nodes (Figure 9). 

In addition to the three known ferroverdins (A, B, and C), 38 other ferroverdin-like molecules were identified by MN (Figure 9). An important heterogeneity was observed between the two strains, both regarding the diversity and the production levels of the identified compounds. It has to be noted that, even if strain MM37 is the strongest ferroverdin A producer of all *S. lunaelactis* strains (Figure 8), the diversity of ferroverdin-like compounds produced is higher in the type strain MM109 with 22 molecules produced only by MM109 and 11 molecules produced only by MM37. Eight ferroverdins, including the three known ferroverdins (A, B and C) are produced by both *S. lunaelactis* strains (Figure 9).

## 4. Discussion

The results described in this work revealed important alterations of both the diversity and the production levels of ferroverdins and bagremycin-related compounds amongst a series of strains belonging to the same species *S. lunaelactis*. If our investigations on the metabolites produced by the cave-dwelling species *S. lunaelactis* would have been limited to the type strain MM109^T^, many new bagremycins and new ferroverdins would not have been discovered. New ferroverdins including ferroverdin D are produced in much higher amounts than the previously known ferroverdin B and C and this is particularly true in strain MM37. This illustrates how the use of more than one strain from a single species allows us to largely increase the panel of compounds produced by a BGC. 

If major distinctions are observed on compounds produced from one single cluster, we obviously expect to observe much more dissimilar metabolite profiles in different strains of the same species if all compounds emanating from their metabolism are included in the comparative analysis. Indeed, mass spectrometry-based metabolite profiling of eleven *S. lunaelactis* strains revealed a series of signals associated with compounds belonging to the same family— and probably originating from the same BGC—displaying contrasting patterns between the strains (Figure 10). Interestingly, chemical profiling of *S. lunaelactis* strains MM22 and MM78, considered to be identical based on average nucleotide identity percentages (Appendix A) or analysis of the ~19,000 nt of the *bag*/*fev* BGC (Figure 3 and Figure 4), revealed that they do not cluster together (Figure 10). This result further demonstrates that “identical” strains does not automatically imply identical specialized metabolism.

Strikingly, 11 clusters of compounds (upper part of the heatmap of Figure 10) are produced by only one of the 11 selected strains of *S. lunaelactis*. This statement suggests that the specialized metabolism of each strain has evolved to optimize the production of one single group of compounds (at least in the culture condition tested). This is possibly an adaptation to life in highly oligotrophic and mineral environments where these strains have been isolated and where the diversity is known to explode (the paradox of the plankton: a limited range of resources supports an unexpectedly wide range of species [37]). The nutrient supply in moonmilk deposits (primarily nitrogen sources [38]) mainly originates from the surface water percolating on the cave walls. The composition of the nutrients in solution varies according to the seasons, the nature of the soil(s) at the surface, the human/animal activities, and the length of the water course before reaching the carbonate deposit. Moonmilk bacterial communities could therefore never reach an equilibrium for which a single species/strain is favored leaving the opportunity of multiple strains of the same species to coexist, each one ready for its optimal conditions to arrive. However, this assumption is certainly not exclusive to cave carbonate speleothems, and can be extended to most environmental niches, thereby emphasizing the advantage of isolating multiple strains of the same species in the context of natural product discovery.

## 5. Conclusions

Our work is the first study that assessed, both at genomic and metabolomic levels, the extent to which different strains that unambiguously belong to the same species share a (dis)similar predisposition for bioactive compound production. Our results clearly demonstrate that taxonomically “identical” strains do not produce identical bioactive metabolites. Differences have been observed both at the production level and in the diversity of molecules produced by the same BGC. Taking the example of bagremycin production, the type strain MM109^T^ is only the best producing strain for bagrelactone, while all other bagremycin-like compounds are produced by other *S. lunaelactis* strains much more. Thus, strain prioritization can certainly prevent the utilization of the best naturally engineered producers. Regarding the new bagremycin-like compounds identified in this study, only 30% were identified in archetype strain MM109^T^, further indicating how strain prioritization could prevent biomolecule discovery. On a larger scale, genome mining of all 18 *S. lunaelactis* strains showed that most BGCs (54%) are strain specific and a chemical profiling analysis further confirmed the high metabolic heterogeneity, even between the closest possible strains (MM22 and MM78 or MM113 and MM115, see Appendix A). In conclusion, our work illustrates how much trying to avoid strain redundancy-based phylogeny can compromise natural product discovery.

## Figures and Tables

**Figure 1 biomolecules-10-01027-f001:**
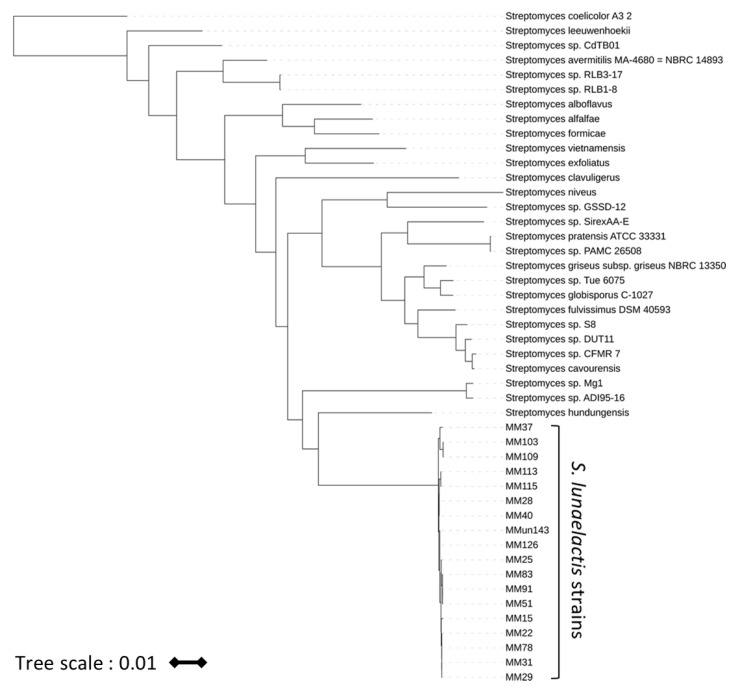
Maximum likelihood phylogenetic tree of *S. lunaelactis* strains and their closest *Streptomyces* counterparts. The tree was generated on concatenated sequences of *atpD-gyrB-recA-rpoB-trpB*. The tree is rooted on *Streptomyces coelicolor*, used as the outgroup, and the tree bar indicates 1% of the estimated sequence divergence.

**Figure 2 biomolecules-10-01027-f002:**
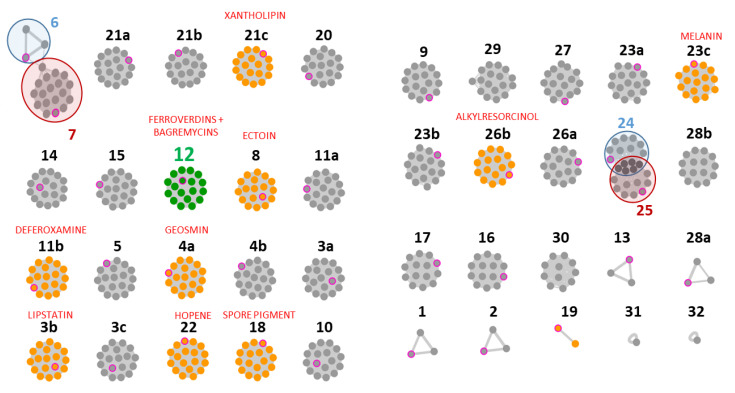
Strain distribution of the 42 biosynthetic gene clusters (BGCs) identified in the genomes of 18 *S. lunaelactis* strains. Each node corresponds to a BGC in one strain, and groups of nodes (“grapes”) indicate that the BGC is conserved in different *S. lunaelactis* strains. The nodes are linked by a grey line whose width is proportional to the orthology level between the connected BGCs. Color code: pink circles indicate nodes of the type strains MM109^T^; green nodes for the *bag*/*fev* BGC; orange nodes refer to known metabolites/BGCs; grey nodes refer to cryptic metabolites/BGCs.

**Figure 3 biomolecules-10-01027-f003:**
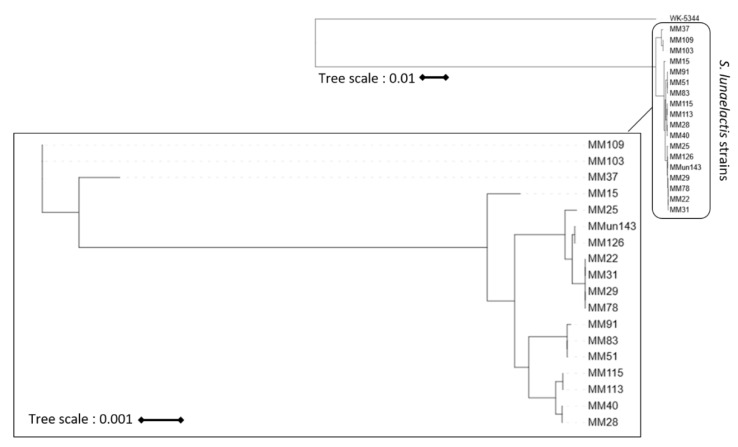
Phylogeny analysis of 18 *S. lunaelactis* strains based on the *fev*/*bag* BGC. The first tree (top right) is rooted on the sequence of the *bag*/*fev* cluster of *Streptomyces* sp. WK-5344 (AB689797.1). The scale bar indicates 1% of estimated sequence divergence. The second tree only includes the 18 *S. lunaelactis* strains to optimally visualize strain clustering. The scale bar indicates 0.1% of estimated sequence divergence.

**Figure 4 biomolecules-10-01027-f004:**
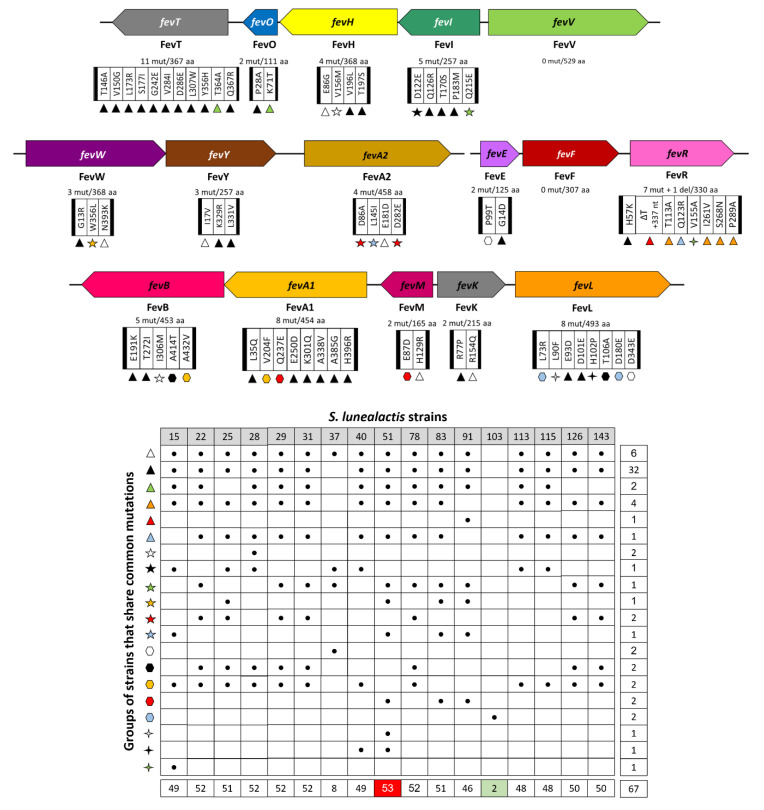
Mutations identified in genes of the *bag*/*fev* BGC in 17 *S. lunaelactis* strains. Top: illustration of the 16 genes of the *bag*/*fev* BGC and localization of the 66 mutations identified. Bottom table: symbols refer to the different groups of *S. lunaelactis* strains that share common mutations. Numbers in the bottom row: total number of mutations per strain compared to the type strain MM109T; Numbers in the right column: number of mutations associated with each symbol.

**Figure 5 biomolecules-10-01027-f005:**
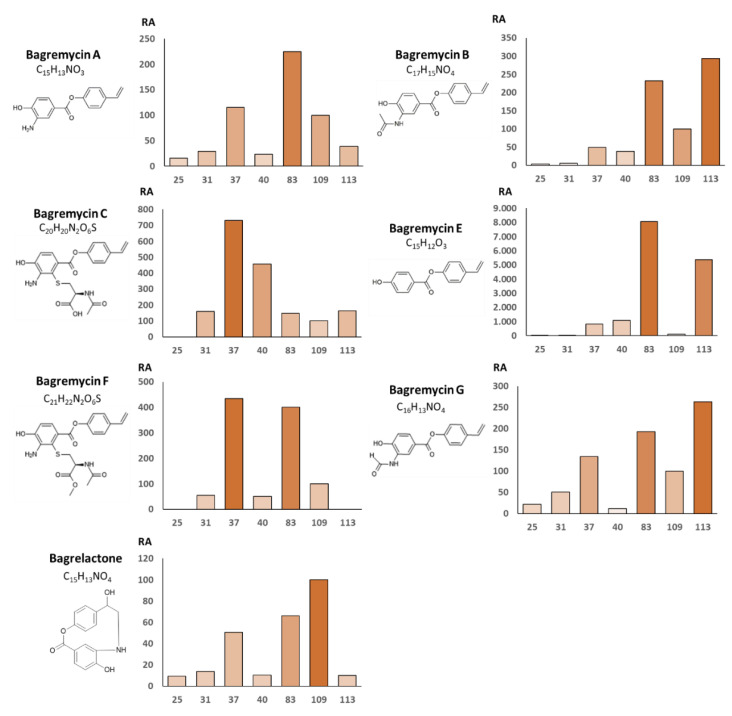
Production levels of bagremycins by seven *S. lunaelactis* strains. The relative abundance (RA) of bagremycins in the full extracts of *S. lunaelactis* strains was performed by peak integration of the extracted ion chromatogram (EIC) using the level produced by *S. lunaelactis* MM109^T^ (fixed to 100%) as reference.

**Figure 6 biomolecules-10-01027-f006:**
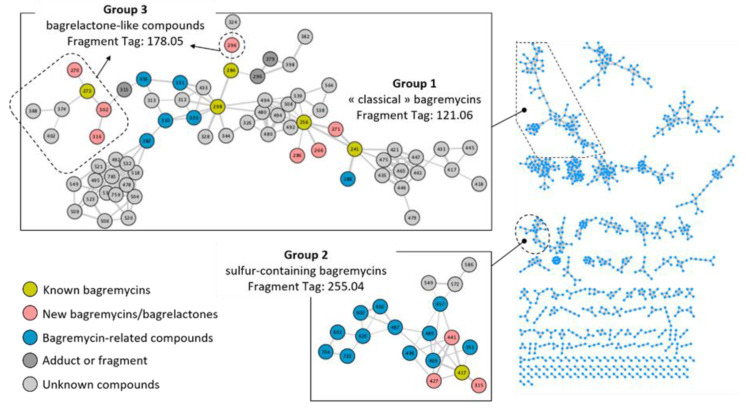
Molecular network of compounds produced by *S. lunaelactis* strains MM25, 31, 37, 40, 83, 109^T^, 113 grown on the ISP7 media. Constellations of nodes were generated using the Global Natural Product Social Molecular Networking (GNPS) software processing the Ultra-Performance Liquid Chromatography–Tandem Mass Spectrometry (UPLC–MS/MS) data obtained for the full extracts of each strain. Each node represents a parent mass (MS1) of a compound detected in the full extract. Compounds are linked by a straight line when they share similar fragmentation patterns. The color code reflects the current knowledge on the structure of a compound: green, previously known bagremycins; pink, new bagremycins identified in this study for which we could propose a structure based on UPLC–MS/MS data; blue, compounds for which we could predict a molecular formula; grey, unknown compounds.

**Figure 7 biomolecules-10-01027-f007:**
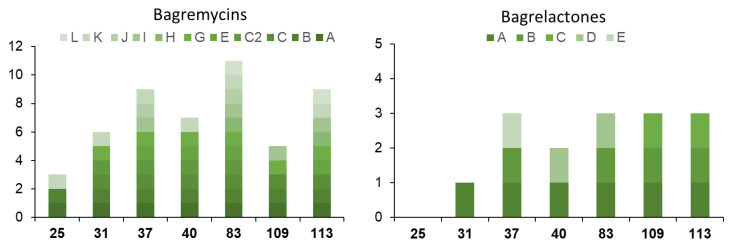
Bagremycins and bagrelactones produced by the seven *S. lunaelactis* strains selected for our Molecular Networking study.

**Figure 8 biomolecules-10-01027-f008:**
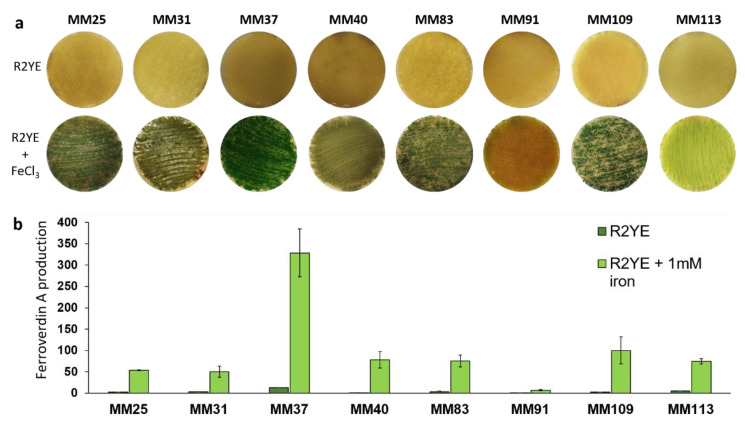
Ferroverdin production in response to iron supply by different strains of *S. lunaelactis*. (**a**) Induced ferroverdin-associated green pigmentation of *S. lunaelactis* strains grown on the R2YE medium upon addition of 1 mM FeCl_3_. (**b**) Semi-quantitative evaluation of ferroverdin A production under conditions non-inducing (R2YE) or triggering (R2YE + 1 mM FeCl_3_) ferroverdin biosynthesis. The production levels were compared to those obtained with strain MM109^T^ (fixed to 100%). Note the overproduction of strains MM37 and the almost complete loss of production for strain MM91.

**Figure 9 biomolecules-10-01027-f009:**
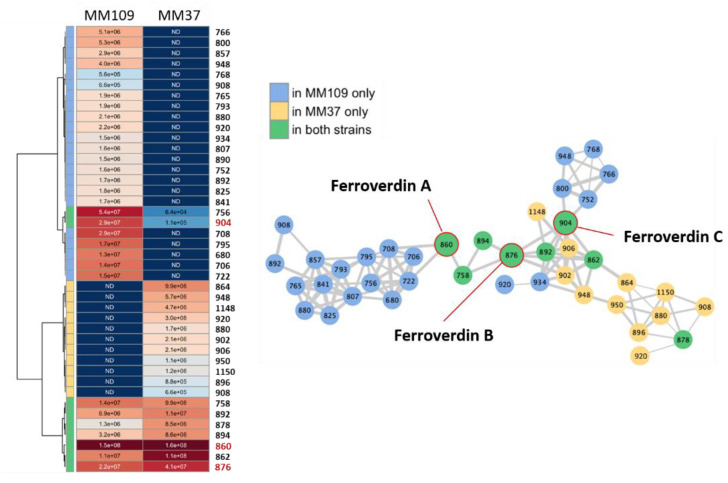
Molecular network and production levels of ferroverdins produced by *S. lunaelactis* strains MM37 and MM109. The molecular network was generated by the GNPS software and manually curated as described in the text. The heatmap on the left reflects the intensity of each of the 41 ferroverdin-like compounds.

**Figure 10 biomolecules-10-01027-f010:**
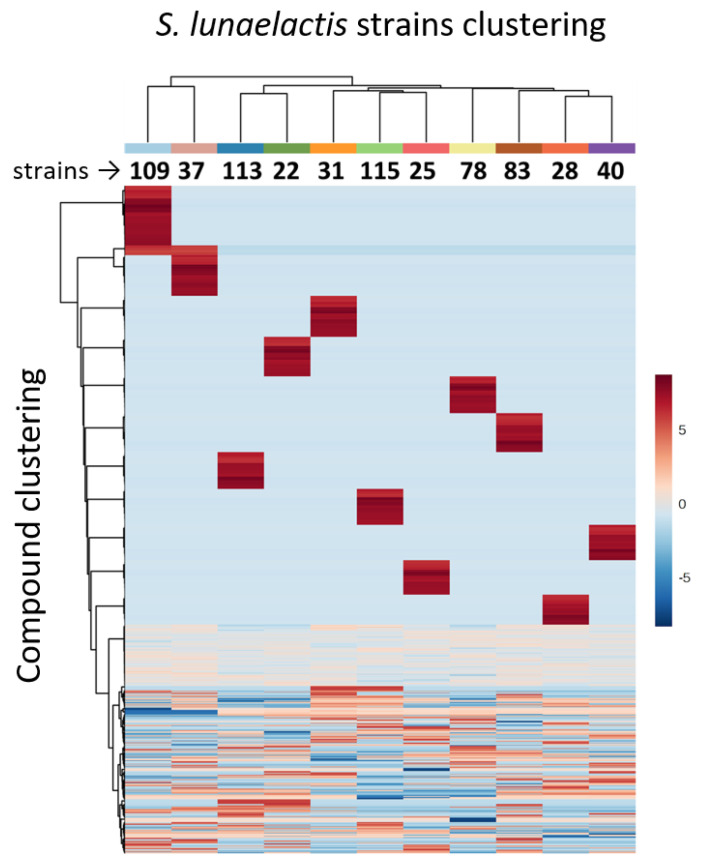
Metabolite profiling of *S. lunaelactis* strains. The heatmap displays all metabolites identified in the crude extracts of 11 S. lunaelactis strains grown on the ISP7 medium. Metabolites are clustered according to their distribution among the strains. The color gradient represents the MS peak intensity (after generalized log normalization and pareto scaling). Columns refer to S. lunaelactis strains, clustered according to their metabolic profiles. Different metabolic cohorts can be identified. Data transformation, clustering, and figures were generated using R package MetaboAnalyst 3.0 [39].

**Table 1 biomolecules-10-01027-t001:** Bagremycin- and bagrelactone-related compounds produced by the different *S. lunaelactis* strains.

#	Name	Molecular Formula	*m*/*z* (exp)	Δm (ppm)	Producing Strains	Group	Refer.
1	Bagremycin A	C_15_H_14_NO_3_^+^	256.0967	0.4	113,25,**31**,83,40, 37,109	1	[34]
2	Bagremycin B	C_17_H_14_NO_4_^+^	298.1074	0.1	**113**,25,31,83,40, 37,109	1	[34]
3	Bagremycin C	C_20_H_20_N_2_O_6_S^+^	417.1115	0.1	113,31,83,40,**37**, 109	2	[35]
4	Bagremycin E	C_15_H_12_O_3_^+^	241.0859	0.3	113,**83**,40,37	1	[35]
5	Bagremycin G	C_16_H_13_NO_4_^+^	284.0915	0.8	**113**,31,83,40,37, 109	1	[35]
6	Bagremycin H	C_16_H_14_O_4_^+^	271.0964	0.2	113,**83**	1 (NL)	This work
7	Bagremycin I	C_16_H_11_NO_3_^+^	266.0812	0.1	113,**83**,37,109	1 (NL)	This study
8	Bagremycin J	C_15_H_12_NO_3_S^+^	286.0531	0.4	83,**37**	1 (NL)	This study
9	Bagremycin C2	C_11_H_8_N_2_O_6_S^+^	315.0643	0.7	31,**83**,40,37	2	This study
10	Bagremycin K	C_21_H_18_N_2_O_6_S^+^	427.0959	0.3	113,25,**31**,83,40,37	ND	This study
11	Bagremycin L	C_22_H_20_N_2_O_6_S^+^	441.1118	0.5	113,**83**	ND	This study
12	Bagrelactone A	C_15_H_13_NO_4_^+^	272.0922	0.2	113,31,83,40,37, **109**	3	[35]
13	Bagrelactone B	C_17_H_13_NO_4_^+^	296.0916	0.2	**83**,37,109,113	ND *	This study
14	Bagrelactone C	C_16_H_15_NO_3_^+^	270.1125	0.1	**113**,**109**	ND *	This study
15	Bagrelactone D	C_16_H_15_NO_5_^+^	302.1023	0.1	**83**,40	3	This study
16	Bagrelactone E	C_17_H_17_NO_5_^+^	316.1179	0.3	**37**	3	This study
From compounds 17 to 33, MS/MS fragmentation did not allow us to predict the structure.							
17		C_19_H_18_NO_6_^+^	356.1127	0.5	37,**109**	3	This study
18		C_15_H_20_NO_6_^+^	310.1285	0.1	25,83,**37**,109	3	This study
19		C_16_H_15_N_2_O_6_^+^	331.0919	0.8	113,25,**40**	3	This study
20		C_17_H_16_NO_6_^+^	330.0969	0.8	25,83,40,37,**109**	3	This study
21		C_13_H_16_NO_6_^+^	282.0972	0.1	113,25,31,83,**37**, 109	3	This study
22		C_16_H_21_ N_2_O_3_^+^	289.1546	0.3	**109**	1 (NL)	This study
23		C_28_H_27_N_2_O_8_S ^+^	551.1484	0.3	**83**	2	This study
24		C_23_H_21_N_2_O_7_S ^+^	469.1066	0.4	**113**,83	ND	This study
25		C_24_H_23_N_2_O_8_S ^+^	499.1170	0.1	113,25,**31**,83,109	ND	This study
26		C_23_H_25_N_2_O_8_S ^+^	489.1328	0.3	113,31,**83**,40,109	2	This study
27		C_22_H_21_N_2_O_7_S^+^	457.1063	0.2	113,31,**83**,40,37, 109	ND	This study
28		C_23_H_23_N_2_O_8_S ^+^	487.1171	0.3	113,**83**	ND	This study
29		C_31_H_28_N_3_O_8_S ^+^	602.1587	0.7	25,**31**,37,109, 113,83	2	This study
30		C_30_H_23_N_3_O_9_S ^+^	602.1231	0.6	113,**83**	ND	This study
31		C_30_H_25_N_3_O_10_S ^+^	620.1336	0.1	31,37,40,83,109, **113**	2	This study
32		C_38_H_32_N_3_O_10_S ^+^	722.1799	0.5	113,**83**	2	This study
33		C_38_H_30_N_3_O_9_S ^+^	704.1696	0.2	113,**83**	2	This study

Lines in grey refers to compounds previously known. For each compound, the best producing strain is highlighted in bold. Abbreviations: mass delta (Δm) in parts per million (ppm) calculated based on the theoretical and the experimental masses and using the Xcalibur software v2.2 software (Thermo Fisher Scientific, San Jose, CA, USA); experimental (exp); neutral loss indicates an alternative non-charged fragmentation of the predicted tag fragment (NL); group not defined (ND); * indicates the presence of a fragment of 176.0706 corresponding to an unsaturated form of the predicted fragment tag of Group 3.

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
