# Peer review of "On the Risks of Phylogeny-Based Strain Prioritization for Drug Discovery: Streptomyces lunaelactis as a Case Study"

_biomolecules, 2020, doi:10.3390/biom10071027_

Round 1

Reviewer 1 Report

The authors studied the genomes and specialized metabolite production of several Streptomyces strains belonging to the species Streptomyces lunaelactis. Their findings illustrate the fact that related strains belonging to the same species (and, moreover, isolated from the same environment in this case) could significantly differ in their ability to produce specialized metabolites. This idea is not new but is very well exemplified by this work in which the authors studied not only the genetic potential of the strains to produce various specialized metabolites (diversity of the specialized metabolites biosynthetic gene clusters) but also the production levels and the chemical diversity for compounds belonging to the family of bagremycins/bagrelactones or ferrovedins. This led to the discovery of several new members of these families of compounds.The experiment design is sound, the conclusions drawn justified and the results are generally clearly and concisely presented. I have only a few comments or questions on relatively minor points.

Specific comments and questions.

  1. In the introduction, the authors cite ref. 8-10 to support the fact that “taxonomically identical species do not display identical metabolomics patterns”. They might also cite Tidjani et al. 2019 (Massive Gene Flux Drives Genome Diversity between Sympatric Streptomyces mBio. 2019 10:e01533-19) which reports that closely related sympatric Streptomyces strains, isolated at a spatial microscale, differ in their specialized metabolite biosynthetic gene clusters and in their antibacterial activities.
  2. The authors might calculate the ANI values for all the S. lunaelactis strain pairs. It might be interesting to have these values in a supplementary table.
  3. In figure 4, the legend for the bottom table might provide information for the figures in the bottom and right part of the table.
  4. Bagremycin production was studied for seven strains (qualified as “representative strains” in the legend of figure 5). How these strains were chosen?
  5. Ferroverdin production was studied for eight strains, the seven for which bagremycin production was studied plus strain MM91. Why this additional strain? Is it because it was almost not producing?
  6. In the discussion, metabolite profiling of eleven strains is discussed. Could this information about metabolite profiling be presented with more details in the Results section?

Typos, etc

  • “deferoxamine” rather than “deferoxamin” (“deferoxamine” in Figure 2 and “deferoxamin” in the text).
  • In figures, the strains are sometimes designated by MM followed by a number ( g. Fig. 8), sometimes by the number only (e. g. Fig. 7).
  • In the references: species names in italics and format of ref. 33.
  • In the text and in table 1, in several cases the reference number is missing and replaced by [ref]

Reviewer 2 Report

This is a really intereting and meaningful research, which can help people to understand the difference of secondary metabolites among the different strains more clearly. This can provide some inspirations for the microbial natrual product researchers. Natural products of microorganisms are strain-specific, which was proved in this manuscript. So, I think this manuscript can be accepted for publication.

Reviewer 3 Report

The work is interesting, the results are well presented. In my opinion, work should be published in Biomolecules.
